# Improving PD-1 blockade plus chemotherapy for complete remission of lung cancer by nanoPDLIM2

Fan Sun[1], Pengrong Yan[1], Yadong Xiao[1,2,3,4], Hongqiao Zhang[3], Steven D Shapiro[2,4], Gutian Xiao[1,3]*, Zhaoxia Qu[1,3]*

[1]UPMC Hillman Cancer Center, Department of Microbiology and Molecular Genetics, University of Pittsburgh School of Medicine, Pittsburgh, United States; [2]Division of Pulmonary, Allergy, and Critical Care Medicine, Department of Medicine, University of Pittsburgh School of Medicine, Pittsburgh, United States; [3]Norris Comprehensive Cancer Center, Hastings Center for Pulmonary Research, Department of Molecular Microbiology and Immunology, University of Southern California Keck School of Medicine, Los Angeles, United States; [4]Department of Medicine, University of Southern California Keck School of Medicine, Los Angeles, United States

## eLife assessment

This study presents a **valuable** finding for the immunotherapy of cancer. The data support the role of PDLIM2 as a tumor suppressor, and more immediately, its relevance for strategies to improve the efficacy of immunotherapy. The evidence supporting the conclusions is **compelling** and the work will be of interest to biomedical scientists working on cancer immunology.

*For correspondence:
Gutian.Xiao@med.usc.edu (GX);
Zhaoxia.Qu@med.usc.edu (ZQ)

Competing interest: The authors declare that no competing interests exist.

**Abstract** Immune checkpoint inhibitors (ICIs) and their combination with other therapies such as chemotherapy, fail in most cancer patients. We previously identified the PDZ-LIM domain-containing protein 2 (PDLIM2) as a bona fide tumor suppressor that is repressed in lung cancer to drive cancer and its chemo and immunotherapy resistance, suggesting a new target for lung cancer therapy improvement. In this study, human clinical samples and data were used to investigate *PDLIM2* genetic and epigenetic changes in lung cancer. Using an endogenous mouse lung cancer model faithfully recapitulating refractory human lung cancer and a clinically feasible nano-delivery system, we investigated the therapeutic efficacy, action mechanism, and safety of systemically administrated PDLIM2 expression plasmids encapsulated in nanoparticles (nanoPDLIM2) and its combination with PD-1 antibody and chemotherapeutic drugs. Our analysis indicate that PDLIM2 repression in human lung cancer involves both genetic deletion and epigenetic alteration. NanoPDLIM2 showed low toxicity, high tumor specificity, antitumor activity, and greatly improved the efficacy of anti-PD-1 and chemotherapeutic drugs, with complete tumor remission in most mice and substantial tumor reduction in the remaining mice by their triple combination. Mechanistically, nanoPDLIM2 increased major histocompatibility complex class I (MHC-I) expression, suppressed multi-drug resistance 1 (MDR1) induction and survival genes and other tumor-related genes expression in tumor cells, and enhanced lymphocyte tumor infiltration, turning the cold tumors hot and sensitive to ICIs and rendering them vulnerable to chemotherapeutic drugs and activated tumor-infiltrating lymphocytes (TILs) including those unleashed by ICIs. These studies established a clinically applicable PDLIM2-based combination therapy with great efficacy for lung cancer and possibly other cold cancers.

**eLife digest** Lung cancer remains the leading cause of all cancer-related deaths. Treatment options are limited because drug-based therapies including chemotherapy and immune checkpoint inhibitors (or ICIs, for short) are ineffective in most patients.

PDLIM2 is a protein that normally prevents tumors from forming by regulating the activities of other genes. However, lung cancer cells generally have lower levels of this protein than healthy cells and this appears to be linked to the ability of the cancer cells to become resistant to chemotherapy and ICIs. Cells make proteins using templates encoded in our DNA. It remains unclear how PDLIM2 production is repressed in lung cancer: it is possible that cancer cells may acquire genetic alterations that affect PDLIM2 production, or there may be other changes to the structure of the DNA known as epigenetic changes.

Sun et al. investigated the production of PDLIM2 in samples from human lung cancer patients. The experiments found that in over 90% of the patients, the levels of PDLIM2 were lower than in cells from healthy individuals. This was due to genetic alterations or epigenetic changes, or a combination of the two.

Further experiments in a mouse model of lung cancer demonstrated that it is possible to use nano-technology to deliver PDLIM2 to cancer cells for effective cancer therapy with low toxicity. Combining this nanotechnology (known as nanoPDLIM2) with both ICIs and chemotherapy drugs was able to completely eradicate all tumors in most of the mice.

The findings provide a firm basis for further studies of the potential of nanoPDLIM2 as a safe and effective therapy for human lung cancer. PDLIM2 production is also repressed in numerous other types of cancer, so it is possible that nanoPDLIM2 may have broader uses in cancer treatment.

## Introduction

Lung cancer is the leading cause of cancer-related deaths in both men and women with a 5-year survival rate of only 22% (*Siegel et al., 2022*). Although these outcomes may be improved by immune checkpoint blockade therapy involving the disruption of the binding of programmed cell death 1 (PD-1, also known as CD279) on tumor-infiltrating lymphocytes (TILs) to programmed death receptor ligand 1 (PD-L1, also known as B7-H1 or CD274) on tumor and tumor-associated cells, most lung cancer patients still fail the therapy, with a response rate of only about 20% (*Doroshow et al., 2019*). In general, this revolutionary immunotherapy works better against 'hot' tumors, which have abundant TILs, strong immunogenicity and sufficient PD-L1 expression. Unfortunately, most tumors are 'cold', with low T-cell infiltration, weak immunogenicity and minimal PD-L1 expression, and show weak response to immune checkpoint inhibitors (ICIs; *Zou et al., 2016*; *Zappasodi et al., 2018*).

Cold lung tumors without targetable oncogenic drivers are treated with chemotherapy as the standard approach (*Baxevanos and Mountzios, 2018*). However, the response rate to this conventional cancer therapy is also low and resistance often occurs after an initial response, with an overall survival (OS) of about 12–17 months. Given its roles in inducing TILs and immunogenicity, in particular PD-L1 expression, to turn cold tumors hot, chemotherapy can be an ideal candidate for combination with PD-1/PD-L1 blockade to improve therapeutic efficacy (*Sun et al., 2019*). Indeed, combination treatment with ICIs and chemotherapeutic drugs shows synergy and better efficacy in both preclinical animal models and clinical trials of lung cancer and other cancers (*Sun et al., 2019*; *Garassino et al., 2020*; *Leonetti et al., 2019*). However, even with the combination of ICIs and chemotherapy, tumors in animals do not remit completely and the objective response rate (ORR) of lung cancer patients only reaches 33–49.7%, with a median progression-free survival (PFS) of just 5.1–9 months and a median OS of 13–22 months (*Garassino et al., 2020*; *Leonetti et al., 2019*). Thus, further improvement over the chemo-immunotherapy is direly needed.

Our recent human and mouse studies have shown that most lung tumors not only have low TILs and decreased PD-L1, but also down-regulate major histocompatibility complex class I (MHC-I), evading recognition and attack by CD8[+] T cells, including those unleashed by ICIs and/or recruited by chemotherapy (*Sun et al., 2019*; *Sun et al., 2020*; *Guo and Qu, 2021*). Following our previous cell line studies (*Qu et al., 2010a*; *Qu et al., 2010b*; *Sun et al., 2015*; *Yan et al., 2009b*; *Vanoirbeek et al., 2014*; *Yan et al., 2009a*; *Fu et al., 2010*), we have established the PDZ-LIM domain-containing protein PDLIM2,

also known as SLIM or mystique (*Torrado et al., 2004*; *Tanaka et al., 2005*; *Loughran et al., 2005*), as a bona fide tumor suppressor and its repression as a causative driver of lung cancer and resistance to ICIs and chemotherapeutic agents (*Sun et al., 2019*). While *PDLIM2* is epigenetically repressed in human lung cancer, associating with therapeutic resistance and poor prognosis, its global or lung epithelial-specific deletion in mice leads to lung cancer development, chemoresistance, and complete resistance to anti-PD-1 and epigenetic drugs. One most important function of PDLIM2 is to promote the ubiquitination and proteasomal degradation of nuclear signal transducer and activator of transcription 3 (STAT3) and nuclear factor-κB (NF-κB) RelA (also known as p65), two master transcription factors that function as proto-oncogenes in lung and many other cancers (*Sun et al., 2019*; *Guo and Qu, 2021*; *Qu et al., 2010a*; *Qu et al., 2010b*; *Sun et al., 2015*; *Tanaka et al., 2007*; *Qu et al., 2012*; *Zhou et al., 2015*; *Qu et al., 2015*; *Steinbrecher et al., 2008*; *Yu et al., 2009*; *Xiao and Fu, 2011*). PDLIM2 repression in tumor cells thus results in the persistent activation of STAT3 and RelA, leading to MHC-I downregulation and high expression of tumor growth-related genes (*Sun et al., 2019*). It also leads to strong induction of multi-drug resistance 1 (MDR1) for acquired chemo-resistance, as chemotherapy further enhances RelA activation for MDR1 transcription in PDLIM2 deficient tumor cells (*Sun et al., 2019*).

In this study, we examined whether and how PDLIM2 can be targeted to treat lung cancer in a faithful mouse model of human lung cancer. In particular, we tested whether systemic administration of nanoparticle-encapsulated PDLIM2-expression plasmids (nanoPDLIM2) could enhance the efficacy of anti-PD-1 and/or chemotherapeutic drugs. We also examined whether PDLIM2 repression in human lung cancer involves genetic deletion and its relationship with epigenetic silencing. These studies indicate that besides epigenetic repression, loss of heterozygosity (LOH) contributes to *PDLIM2* downregulation in about 58% of human lung tumors, and that *Pdlim2* heterozygous deletion (*Pdlim2*[+/-]) mice develop spontaneous tumors in lung and other organs. Notably, systemic administration of nanoPDLIM2 reverses the phenotypes caused by PDLIM2 repression and induces complete remission of all lung tumors in most mice without further increasing toxicity when combined with anti-PD-1 and chemotherapeutic drugs. These findings provide a firm basis to combine ICIs and chemotherapeutic drugs with PDLIM2-targeted therapy for the treatment of lung and other cancers.

## Results
### Both epigenetic repression and genetic deletion of *PDLIM2* in lung cancer

Using 40% of the expression level in matched normal lung tissues as the cut-off in the analysis of the Cancer Genome Atlas (TCGA) data, we found that *PDLIM2* was repressed in over 75% of human lung tumors in a prior study (*Sun et al., 2019*). If using 50% as the cut-off, *PDLIM2* was repressed in about 94% of human lung tumors (*Figure 1A*). Using 125% of the methylation level of the *PDLIM2* promoter in normal lung tissues as the cut-off to analyze TCGA database, over 70% of human lung tumors were found to have hypermethylation of the *PDLIM2* promoter (*Figure 1A and B*), further validating our previous finding of epigenetic silencing as the main mechanism underlying PDLIM2 repression (*Sun et al., 2019*).

In line with *PDLIM2*'s location on chromosome 8p21.3, a frequent LOH region in lung and other tumors (*Wistuba et al., 1999*; *Kang, 2015*; *Macartney-Coxson et al., 2008*; *Swalwell et al., 2002*; *Wurster et al., 2017*), analysis of TCGA database revealed that *PDLIM2* expression was positively associated with its gene copy numbers, and that over 58% of human lung tumors had genetic deletion of the *PDLIM2* gene if copy number variation (CNV) of –0.1 was used as the cut-off for the gene deletion (*Figure 1C*). Further analysis indicated that about 44% of human lung tumors simultaneously harbored the promoter hypermethylation and LOH of the *PDLIM2* gene, and approximately 27% and 14% of them only having the promoter hypermethylation or LOH of the *PDLIM2* gene, respectively (*Figure 1D*). Around 15% of lung tumors possessed no such epigenetic or genetic alterations in *PDLIM2*. These findings were confirmed by microsatellite and gene-specific PCR-based LOH analysis of human primary lung tumor tissues and cell lines (*Figure 1E and F*). These data suggested that PDLIM2 repression in lung cancer involves both epigenetic silencing and genetic deletion.

To determine the significance of *PDLIM2* LOH in lung tumors, we examined whether *Pdlim2* heterozygous deletion leads to decreased PDLIM2 expression and spontaneous tumors in mice. Unlike its

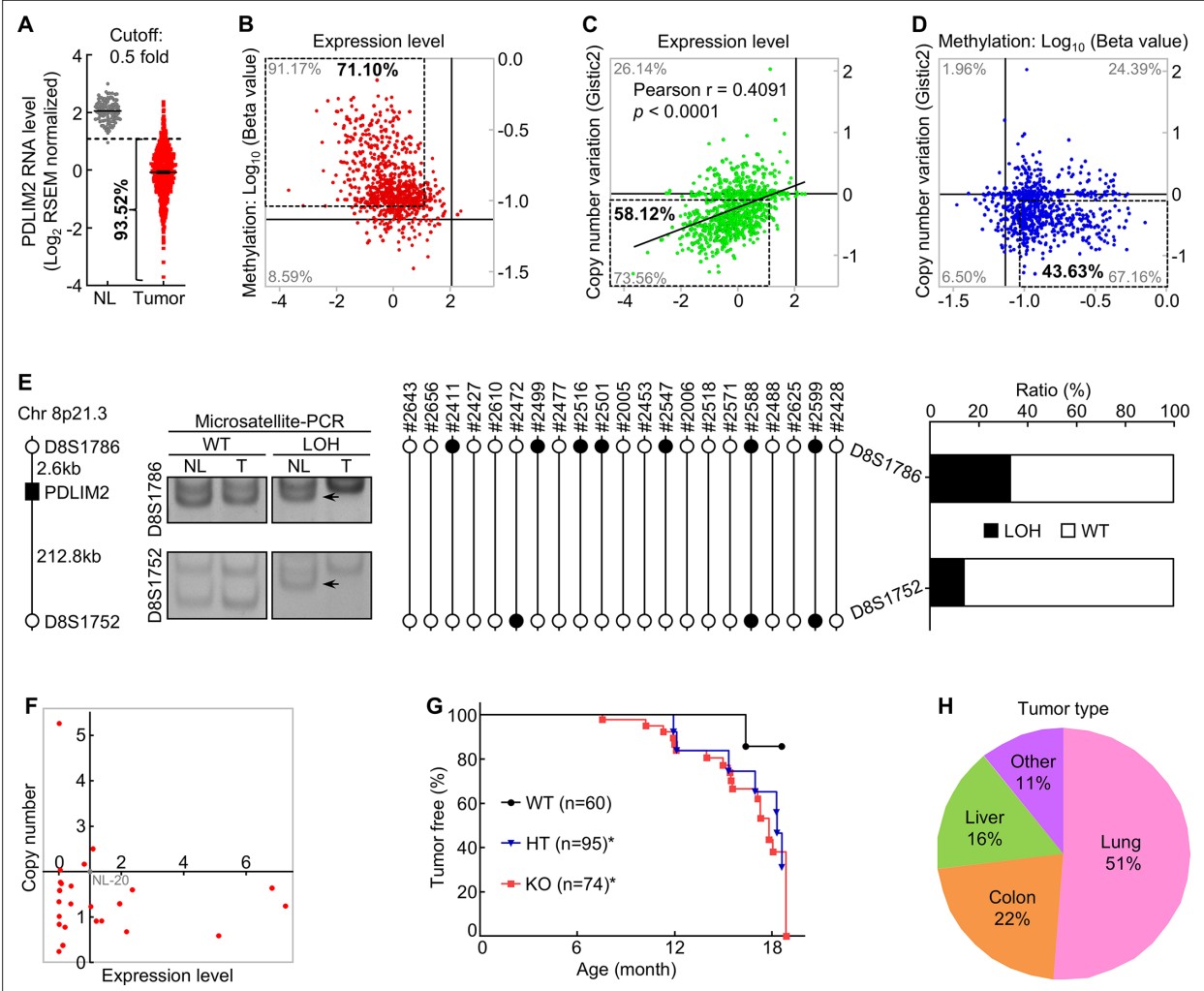

**Figure 1.** PDLIM2 repression in human lung cancer involves both epigenetic alteration and genetic deletion, and *Pdlim2* genetic deletion in mice leads to development of spontaneous tumors, majorly lung tumors. (**A**) TCGA data showing *PDLIM2* repression in over 90% of lung tumors if using 50% of the expression level in normal lung tissues as the cut-off (NL, n=110; Tumor, n=1019). (**B**) TCGA data showing *PDLIM2* promoter hypermethylation and expression repression (dashed box) in over 70% of lung tumors when using 125% of the methylation level in normal lung tissues as the cut-off (n=827). (**C**) TCGA data showing positive associations between *PDLIM2* expression and its gene copy numbers as well as *PDLIM2* genetic deletion and expression repression (dashed box) in about 58% of lung tumors using the copy number variation of –0.1 as the cut-off (n=1010). (**D**) TCGA data showing simultaneous promoter hypermethylation and genomic deletion of *PDLIM2* (dashed box) in about 44% of lung tumors (n=816). (**E**) Microsatellite-PCR showing *PDLIM2* loss in human lung tumors (n=21). (**F**) qPCR showing *PDLIM2* loss in human lung cancer cell lines with known copy number of the *PDLIM2* gene (n=25). (**G**) Kaplan-Meier tumor-free survival curve showing increased spontaneous tumors in *Pdlim2⁻/⁻* and *Pdlim2⁺/⁻* mice compared to WT mice. Gehan-Breslow-Wilcoxon test was performed. *<0.05. (**H**) Percentage of tumor types spontaneously developed in *Pdlim2⁻/⁻* and *Pdlim2⁺/⁻* mice showing a majority of lung tumors.

The online version of this article includes the following source data and figure supplement(s) for figure 1:

**Figure supplement 1.** Mice with PDLIM2 deletion develop spontaneous tumors, majorly lung tumors.

**Source data 1.** Excel file for the data shown in *Figure 1A-G*.

**Source data 2.** Original files for the DNA gel images shown in *Figure 1E*.

**Source data 3.** PDF file for the DNA gel images shown in *Figure 1E* with the relevant bands clearly labelled.

absolute absence in *Pdlim2* homozygous deletion (*Pdlim2⁻/⁻*) mice, PDLIM2 was detected in the lung of *Pdlim2⁺/⁻* mice, but at a much lower level compared to wild type (WT) mice (***Figure 1—figure supplement 1A***). Importantly, *Pdlim2⁺/⁻* mice, like *Pdlim2⁻/⁻* mice, also developed spontaneous tumors (***Figure 1G***). Of note, over 50% of tumors developed in *Pdlim2⁻/⁻* and *Pdlim2⁺/⁻* mice were lung tumors (***Figure 1H***; ***Figure 1—figure supplement 1B***). These data are also highly consistent with the fact

that PDLIM2 is ubiquitously expressed under physiological conditions, with the highest level in the lung and lung epithelial cells in particular (*Sun et al., 2019*; *Torrado et al., 2004*; *Tanaka et al., 2005*; *Loughran et al., 2005*). Thus, PDLIM2 is a haploinsufficient tumor suppressor that is particularly important for lung tumor suppression.

## Efficacy of systemic administration of nanoPDLIM2 in refractory lung cancer

Although reversal of PDLIM2 epigenetic repression by epigenetic drugs to restore PDLIM2 expression in cancer cells may be used to treat lung cancer (*Sun et al., 2019*), it is logical that this approach cannot be applied to lung tumors involving *PDLIM2* LOH, which accounts for about 58% of all lung cancer cases. To overcome this limitation and expand PDLIM2-targeted therapy to all lung tumors with PDLIM2 repression regardless of the involved mechanisms, we tested the therapeutic efficacy of systemic administration of PDLIM2-expression plasmids encapsulated by the clinically feasible in vivo-jetPEI (*Matouk et al., 2013*; *Buscail et al., 2015*; *Nyamay'Antu et al., 2019*; *Bonnet et al.,*

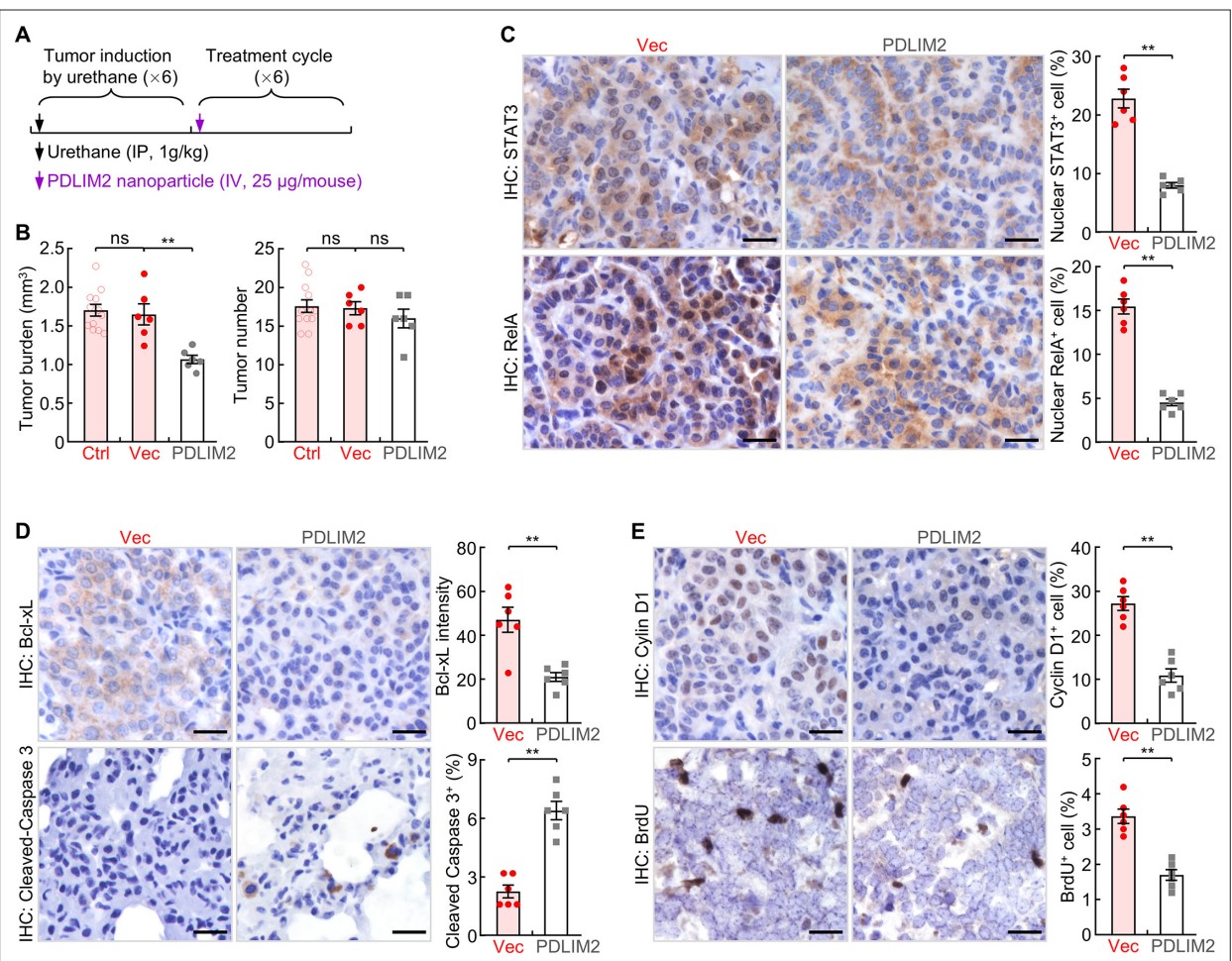

**Figure 2.** Systemic administration of PDLIM2 plasmid nanoparticles shows efficacy in mouse model of refractory lung cancer. (**A**) Schedule of lung cancer induction and treatment. (**B**) Urethane model showing efficacy of intravenous administration of PDLIM2-expression plasmid nanoparticles for refractory lung cancer (n≥6). Nanoparticles with an empty vector plasmid (Vec) that was employed to express PDLIM2 were used as a control. (**C**) IHC staining showing decreased nuclear expression of STAT3 and RelA in lung tumors by PDLIM2 nanotherapy (n=6). (**D**) IHC staining showing decreased Bcl-xL and increased apoptosis marker cleaved caspase –3 in lung tumors by PDLIM2 nanotherapy (n=6). (**E**) IHC staining showing decreased Cyclin D1 and proliferation (BrdU incorporation) in lung tumors by PDLIM2 nanotherapy (n=6). Scale bar in (**C–E**), 20 μm. Student's *t* test was performed (two tailed, unpaired) and data represent means ± SEM in (**B–E**). **p<0.01; ns, not statistically significant.

The online version of this article includes the following source data for figure 2:

**Source data 1.** Excel file for the data shown in *Figure 2B-E*.

2008; *Yang et al., 2013*). To this end, we employed mouse lung tumors induced by urethane, a faithful model of human lung cancer and adenocarcinoma (AC) in particular (*Sun et al., 2019*; *Zhou et al., 2015*; *Qu et al., 2015*; *Kellar et al., 2015*; *Zhou et al., 2017*; *Li et al., 2018*; *Sun et al., 2016*). Urethane is a chemical carcinogen present in fermented food, alcoholic beverage and also cigarette smoke, the predominant risk factor accounting for about 90% of human lung cancer cases (*Hecht, 2002*). Like its human counterpart, the murine lung cancer induced by urethane also shares PDLIM2 repression, in addition to their similarities in histology, genetics, molecular biology, and immunology (*Sun et al., 2019*; *Sun et al., 2020*; *Zhou et al., 2015*; *Qu et al., 2015*; *Kellar et al., 2015*; *Zhou et al., 2017*; *Li et al., 2018*; *Sun et al., 2016*). WT mice with lung tumors induced by urethane were i.v. injected with nanoparticle-encapsulated *Pdlim2* plasmids or empty vector plasmids. Six weeks post the initial treatment of nanotherapy, mice were euthanized, and lung tissues were collected (*Figure 2A*). Compared to the vector plasmid mock-treated group (Vec), nanoPDLIM2-treated mice had a significantly reduced tumor burden in their lungs, although the decrease in tumor numbers was not statistically significant (*Figure 2B*).

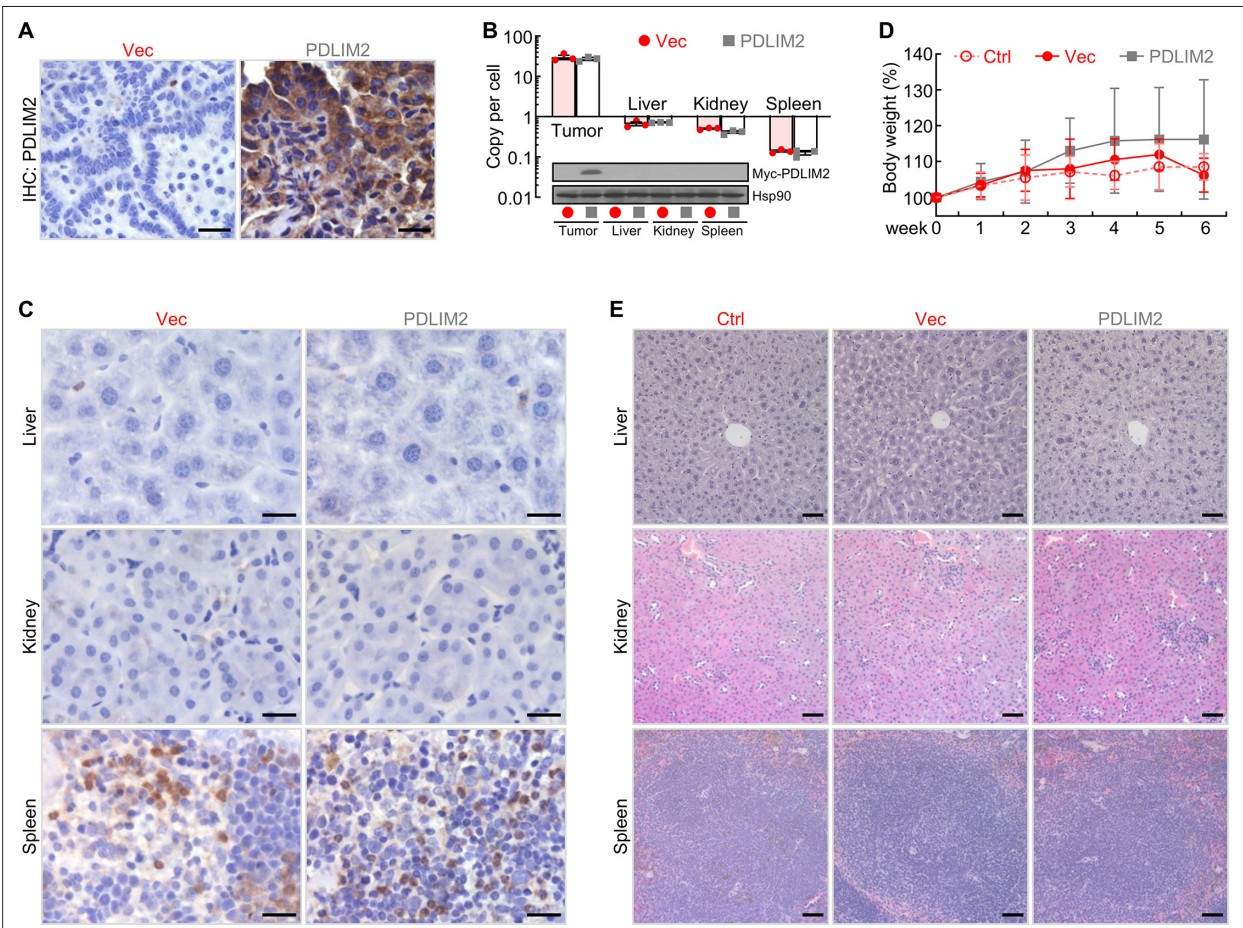

**Figure 3.** PDLIM2 nanotherapy shows high tumor specificity and low toxicity. (**A**) IHC staining showing high PDLIM2 re-expression in lung tumors after PDLIM2 nanotherapy. (**B**) PCR and IB assays showing lung tumor-specific plasmids delivery and PDLIM2 expression by PDLIM2 nanotherapy (n=3). (**C**) IHC staining showing comparable expression of PDLIM2 in the indicated tissues of mice treated with PDLIM2 expression plasmid or empty vector plasmid nanoparticles. (**D**) No significant changes in animal body weight by nanoPDLIM2 (n=5). (**E**) H&E staining showing no noticeable changes in major organs by nanoPDLIM2. Scale bar: (**A and C**) 20 µm; (**E**) 50 µm.

The online version of this article includes the following source data for figure 3:

**Source data 1.** Excel file for the data shown in *Figure 3B, D*.

**Source data 2.** Original files for the Western blot images shown in *Figure 3B*.

**Source data 3.** PDF file for the Western blot images shown in *Figure 3B* with the relevant bands clearly labelled.

In line with the tumor reduction, nanoPDLIM2 administration decreased nuclear RelA and STAT3, a hallmark of NF-κB and STAT3 activation, accordingly reduced the expression of their downstream cell survival gene Bcl-xL and cell proliferation gene Cyclin D1, increased apoptosis and decreased proliferation of lung cancer cells (*Figure 2C-E*). These data indicated that intravenous administration of nanoPDLIM2 shows efficacy in suppressing oncogenic RelA and STAT3 activation and in treating lung cancer.

## High tumor specificity and low toxicity of systemic administration of nanoPDLIM2

To characterize the potential new lung cancer therapy, we examined the expression levels of PDLIM2 in lung tumors and several organs, including the liver, kidney, and spleen. Consistent with the treatment efficacy, a high level of PDLIM2 was detected in the lung tumors from mice treated with the PDLIM2 plasmid nanoparticles one week post nanoPDLIM2 treatment, whereas no obvious PDLIM2 was found in the lung tumors from mice treated with the control plasmid nanoparticles (*Figure 3A and B*). It should be pointed out that the tumor delivery efficiency of PDLIM2-expression or control plasmid nanoparticles was similarly high (*Figure 3B*). However, either PDLIM2-expression or control plasmids were hardly detected in other organs/tissues of the same mice, including liver, kidney and spleen. Of note, in the first two days after injection, both empty vector and PDLIM2-expression plasmids were also detected at high levels in those tissues but quickly cleared afterward (data not shown), consistent with the well-documented tumor-specific enrichment of nanoparticles (*Yang et al., 2013*; *Commisso et al., 2013*; *Huang et al., 2017*; *Shi et al., 2017*). Consistently, ectopic Pdlim2 was not detected, and the levels of PDLIM2 proteins were comparable in these tissues (*Figure 3B and C*).

In further support of the lung tumor-specific delivery, the i.v. injection of either PDLIM2-expression or control plasmid nanoparticles showed no obvious toxicity to animals, as evidenced by no significant changes in the animal body weight and histology of all organs/tissues we examined, including spleen, kidney and liver (*Figure 3D and E*). Mouse appearance and behaviors, such as eating, drinking, defecating, urinating, sniffing, grooming, and digging, were not different between PDLIM2-expression and control plasmid nanoparticle groups (data not shown). Taken together, these data suggested the therapeutic efficacy and low toxicity of PDLIM2-based nanotherapy in the mouse model of refractory lung cancer.

## Synergy of nanoPDLIM2 with chemotherapy in lung cancer treatment

Given the role of PDLIM2 in inhibiting the expression of cell survival and proliferation genes in tumor cells, which contribute to chemoresistance, we tested whether nanoPDLIM2 increases the efficacy of chemotherapy in the mouse model of lung cancer. Treatment with carboplatin and paclitaxel, two chemotherapeutic drugs that are often used together as the first-line treatment for lung and many other cancers, led to significant decrease in tumor number and tumor burden (*Figure 4A*). Importantly, combination with nanoPDLIM2 further significantly decreased both tumor number and tumor burden, suggesting a promising synergy between PDLIM2 nanotherapy and chemotherapy in lung cancer treatment. Consistently, significantly higher tumor cell apoptosis was detected in mice treated with the combination therapy, in comparison to those with nanoPDLIM2 or chemotherapy alone (*Figure 4B*).

Another important mechanism contributing to the synergy between these two therapies involves nanoPDLIM2 blockade of the acquired chemoresistance of lung cancer cells. Chemotherapy induced strong RelA activation/nuclear expression and MDR1 expression (*Figure 4C and D*). NanoPDLIM2 not only repressed the constitutive activation of RelA in cancer cells but also prevented the strong induction of RelA activation and MDR1 expression by the chemotherapy. Thus, PDLIM2 nanotherapy improves the therapeutic efficacy of chemotherapy through blocking both intrinsic and acquired chemoresistance of lung cancer cells.

## NanoPDLIM2 enhancement of ICI's efficacy in lung cancer treatment

High expression of cell survival genes also renders tumor cells resistant to the tumoricidal activity of cytotoxic T lymphocytes (CTLs), including those unleashed by ICIs. PDLIM2 nanotherapy should also enhance the efficacy of immunotherapy, given its ability in suppressing the expression of cell survival genes. Furthermore, NanoPDLIM2 increased the number of TILs and the expression of MHC-I,

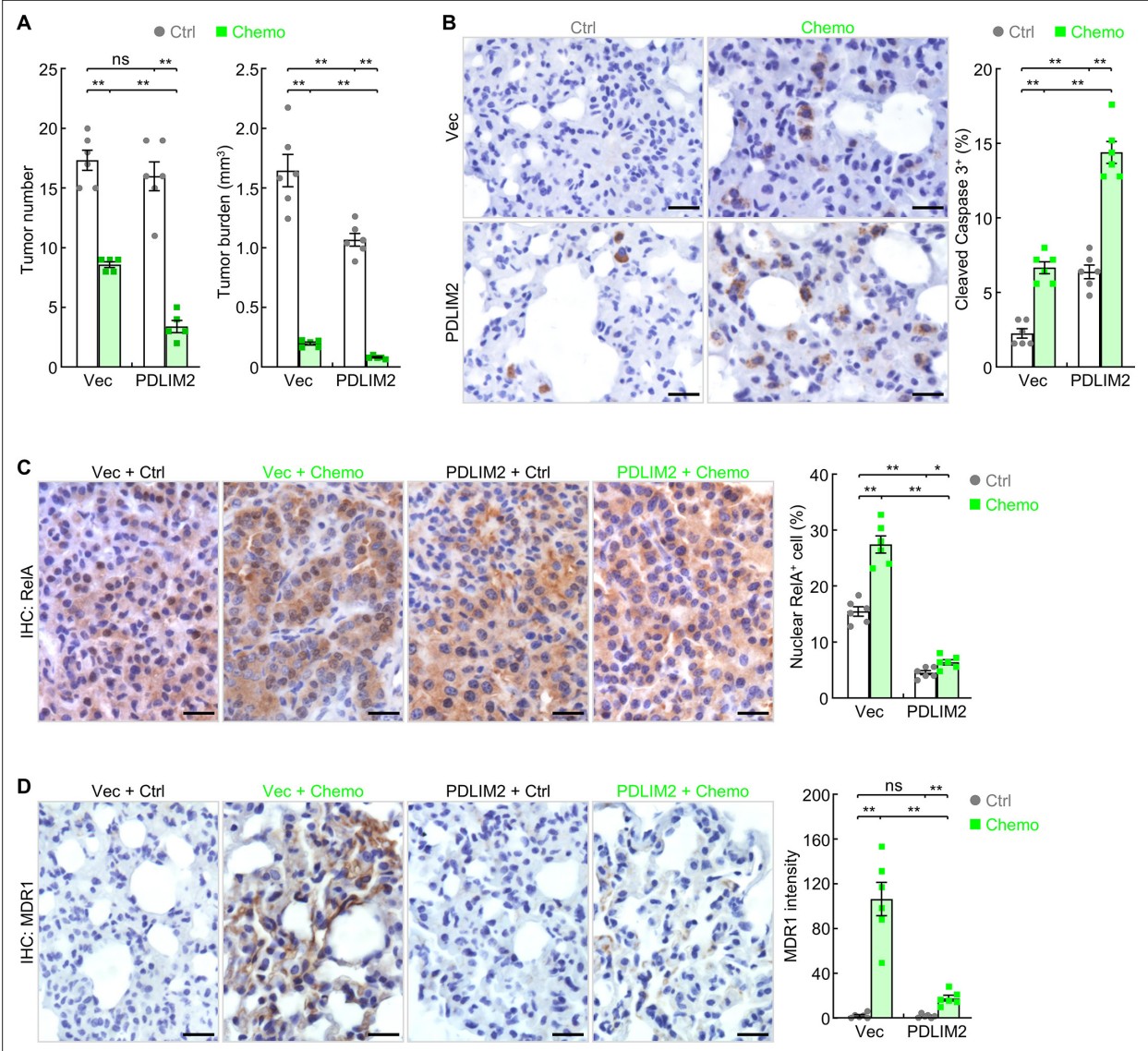

**Figure 4.** PDLIM2 nanotherapy renders lung cancers more vulnerable to chemotherapy. (**A**) Urethane model showing synergy of PDLIM2 nanotherapy and chemotherapy in lung cancer treatment (n≥5). (**B**) IHC staining showing increased lung tumor cell apoptosis by PDLIM2 nanotherapy, chemotherapy, and further increase by their combination (n=6). (**C**) IHC staining showing RelA activation by chemotherapy and blockage of chemo activation of RelA by PDLIM2 nanotherapy (n=6). (**D**) IHC staining showing strong MDR1 induction by chemotherapy and blockage of MDR1 induction by PDLIM2 nanotherapy (n=6). Scale bar in (**B–D**), 20 μm. Student's *t* test was performed (two tailed, unpaired) and data represent means ± SEM. *p<0.05; **p<0.01; ns, not statistically significant.

The online version of this article includes the following source data for figure 4:

**Source data 1.** Excel file for the data shown in *Figure 4A-D*.

the most important and core components of immunotherapies including PD-1 immune checkpoint blockade therapy (*Figure 5A and B*). In line with our previous studies (*Sun et al., 2019*), PD-1 blocking antibody showed some efficacies in the mouse model of lung cancer, as evidenced by the significant decrease in tumor burden and tumor number (*Figure 5C*). Combining with PDLIM2 nanotherapy further significantly decreased tumor burden, although the decrease in tumor number failed to reach statistical significance. Consistently, significant increases in both CD4+ and CD8+ TILs, CD8+ CTL activation and lung tumor cell death were detected (*Figure 5D-F*).

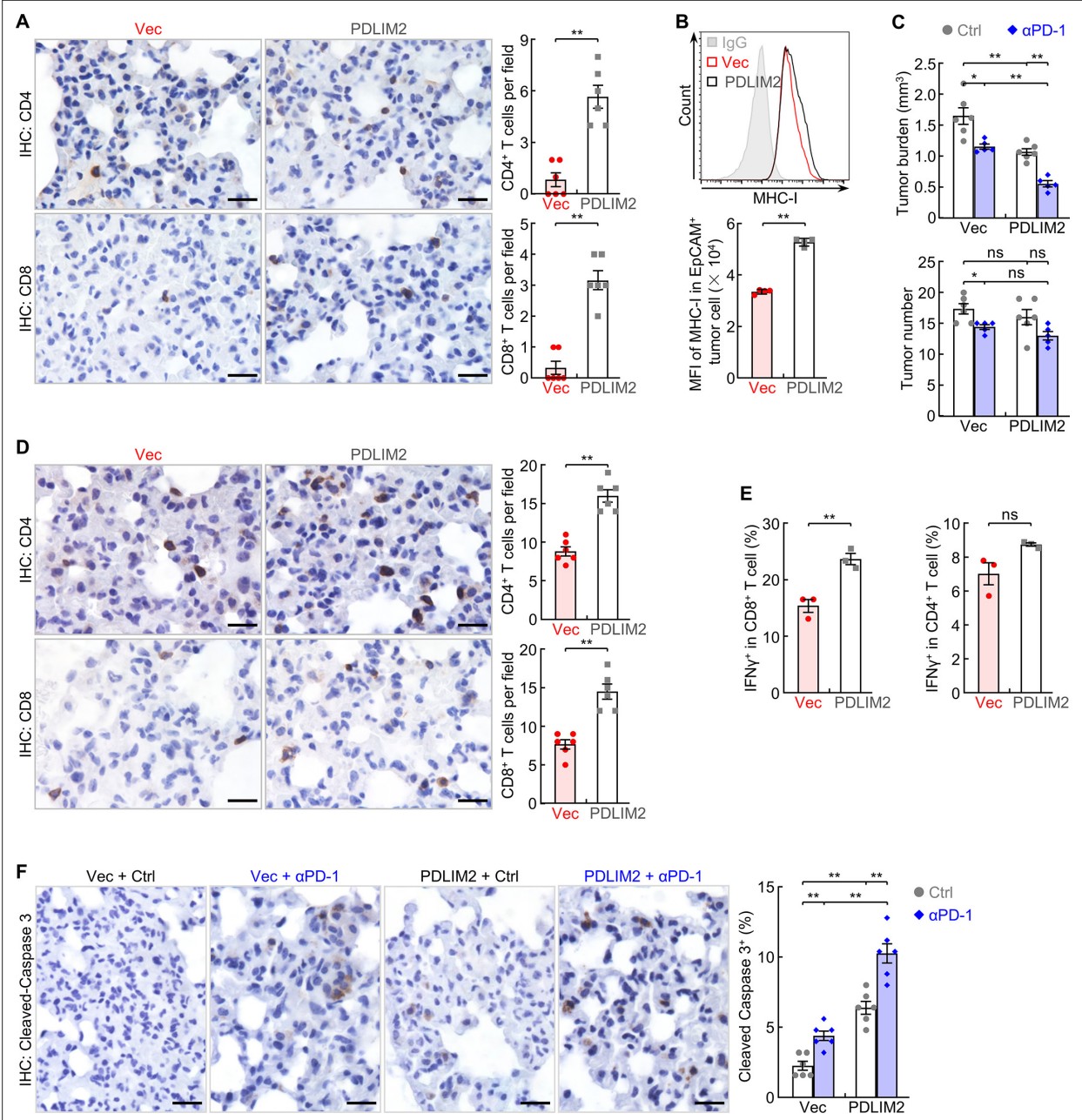

**Figure 5.** PDLIM2 nanotherapy increases the efficacy of PD-1 blockade immunotherapy for refractory lung cancer. (**A**) IHC staining showing increased TILs in lung tumors by PDLIM2 nanotherapy (n=6). (**B**) FACS showing increased MHC-I expression in lung tumor cells by PDLIM2 nanotherapy (n=4). (**C**) Urethane model showing PDLIM2 nanotherapy enhancing PD-1 immunotherapy efficacy in lung cancer treatment (n≥5). (**D**) IHC staining showing increased TILs by PDLIM2 nanotherapy in the context of immunotherapy (n=6). (**E**) FACS showing increased activation of CD8+ T cells by PDLIM2 nanotherapy in the context of immunotherapy (n=3). (**F**) IHC staining showing increased lung tumor cell apoptosis by PDLIM2 nanotherapy, immunotherapy, and further increase by their combination (n=6). Scale bar in (**A, D, F**), 20 μm. Student's *t* test was performed (two tailed, unpaired) and data represent means ± SEM. **p<0.01; ns, not statistically significant.

The online version of this article includes the following source data for figure 5:

**Source data 1.** Excel file for the data shown in *Figure 5A-F*.

## Complete remission of all lung tumors in most mice by the combination treatment of nanoPDLIM2 and anti-PD-1 and chemotherapeutic drugs

Like most human lung tumors, lung tumors in our animal model exhibit decreased expression of PD-L1 on cell surface, in additional to the low expression of MHC-I and low number of TILs (*Sun et al., 2019*;

*Sun et al., 2020*). In line with our previous finding that PD-L1 expression is largely independent of PDLIM2 (*Sun et al., 2019*), nanoPDLIM2 failed to induce PD-L1 expression in lung tumors, which was in contrast to chemotherapy (*Figure 6A*). This may explain why the enhancing effect of nanoPDLIM2 on PD-1 blockade therapy is only moderate.

Although it dramatically increases TILs as well (*Sun et al., 2019*), chemotherapy also induced the expression of PD-L1 on cancer cells (*Figure 6A*), which presumably protects cancer cells from immune attack and thereby restricts further efficacy improvement of its combination with PDLIM2 nanotherapy. But it may increase the sensitivity of cancer cells to PD-1 blockade therapy, particularly in combination with nanoPDLIM2. Indeed, chemotherapy and PD-1 blockade therapy showed a promising synergy in reducing both tumor number and tumor burden in comparison to the individual treatment of chemotherapy or PD-1 blockade therapy (*Sun et al., 2019*). The synergy was largely blocked when *Pdlim2* was genetically deleted from lung cancer cells (ΔSPC; *Figure 6B and C*), suggesting an important role of PDLIM2 in the combination therapy of chemotherapeutic drugs and anti-PD-1. Whereas PDLIM2 nanotherapy increased MHC-I expression on tumor cells (*Figure 5B*), which is critical for better recognition and killing of tumor cells by CD8+ CTLs, chemotherapy alone or its combination with anti-PD-1 failed to do so (*Figure 6D*). Accordingly, combination of anti-PD-1 and chemotherapeutic drugs, like the combination therapies of nanoPDLIM2 and anti-PD-1 or chemotherapeutic drugs, also failed to induce a complete remission of lung tumors in any mice.

Given their overlapping roles in increasing TILs and in particular their complement roles in inducing MHC-I and PD-L1 expression on tumor cells, which turn tumors hot and more sensitive to PD-1 blockade, combination of nanoPDLIM2 and chemotherapeutic drugs with anti-PD-1 is expected to show better efficacy compared to combining only two of them. Indeed, combination of all three showed the strongest effect on reducing tumor burden and numbers, and caused complete remission of all lung tumors in 60% of mice (*Figure 6E*) and much more shrinkage of lung tumors in remaining mice in comparison to those treated with two combined therapies. In line with the high therapeutic efficacy, the triple combination therapy significantly increased the numbers and/or activation of CD4+ and CD8+ T cells in the lung, compared to the combination of chemotherapeutic drugs and anti-PD-1 (*Figure 6F and G*).

Of note, consistent with the undetectable toxicity of PDLIM2 nanotherapy, its co-treatment did not further increase the toxicity of anti-PD-1 and chemotherapeutic drugs. There was no obvious histological difference of major organs, including the liver, lung, kidney, and spleen (*Figure 6—figure supplement 1*). Moreover, no significant differences in animal body weights were observed by additional nanoPDLIM2, in comparison to the mice received the combinational treatment of anti-PD-1 and chemotherapeutic drugs in the presence or absence of empty vector plasmid nanoparticles (*Figure 6H*). These data suggested a novel combination therapy with very high therapeutic efficacy and no increased toxicity for lung cancer, particularly refractory lung cancer.

## Discussion

PD-1/PD-L1 blockade immunotherapy has recently joined chemotherapy as a standard treatment for lung and several other cancers (*Doroshow et al., 2019*; *Zou et al., 2016*; *Zappasodi et al., 2018*). While some patients have shown dramatic responses, most patients do not benefit from this novel treatment. Currently, various combinations of ICIs with other therapies, in particular its combination with chemotherapeutic drugs are being extensively tested in both preclinical and clinical trial studies to expand the benefit of this innovative therapy (*Sun et al., 2019*; *Garassino et al., 2020*; *Leonetti et al., 2019*). Although a promising synergy and better efficacy has been observed in both preclinical animal models and human clinical trial studies (*Sun et al., 2019*; *Garassino et al., 2020*; *Leonetti et al., 2019*), significant further improvement is direly needed. Using an authentic mouse model of lung cancer, we show, for the first time, that PDLIM2 nanotherapy shows efficacy and high safety, and more importantly, induces complete remission of all lung tumors in most animals when it is combined with anti-PD-1 and chemotherapeutic drugs.

Most human lung tumors as well as lung tumors in our animal model have low numbers of TILs and decreased expression of PD-L1 and MHC-I on the cell surface (*Sun et al., 2019*; *Sun et al., 2020*), all of which are important mechanisms contributing to the resistance to PD-1 blockade therapy. Through inducing immunogenic cell death (ICD) of cancer cells (*Leonetti et al., 2019*), chemotherapy can increase TILs and PD-L1 expression on tumor cells, and thereby synergize with anti-PD-1. However,

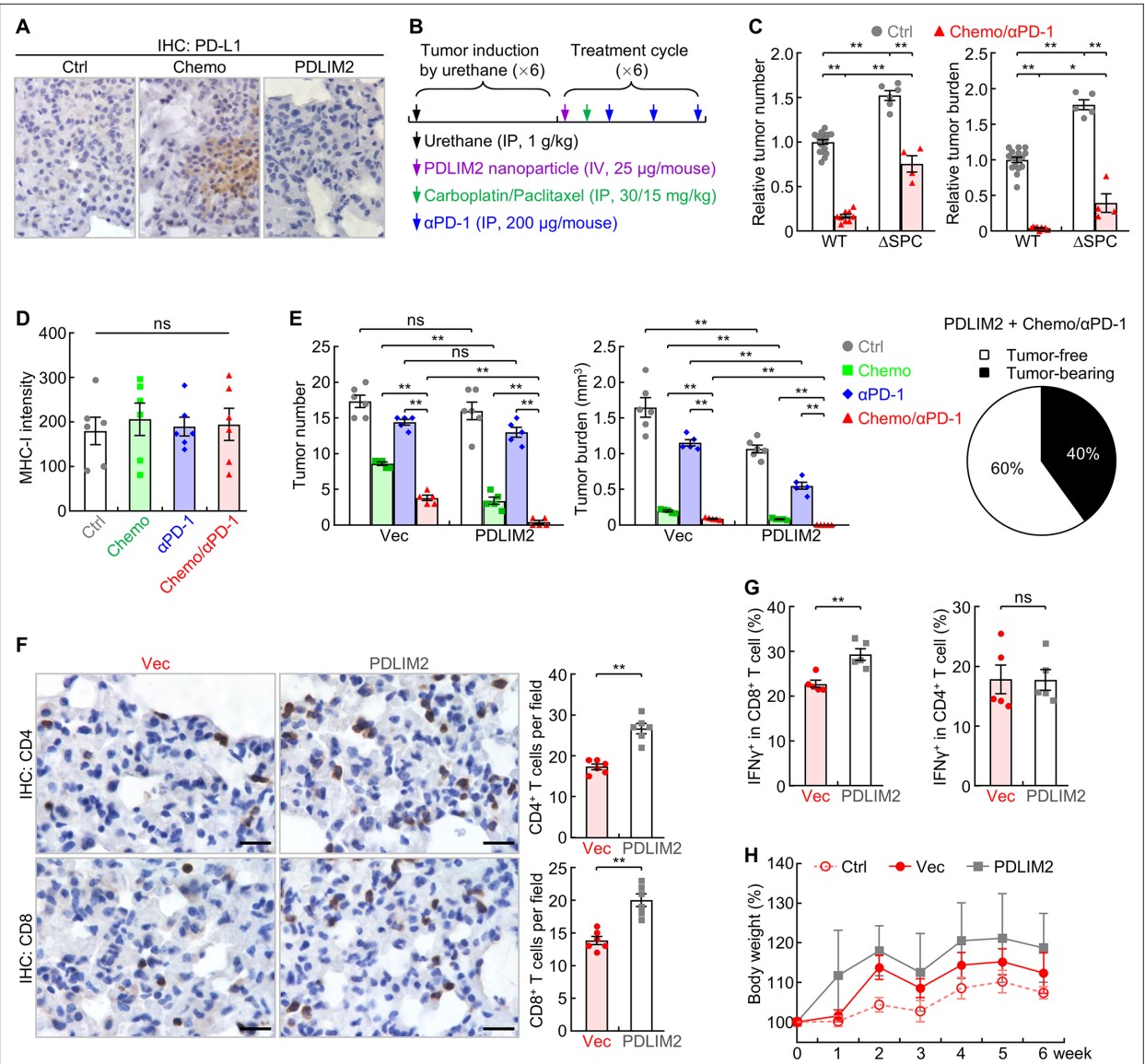

**Figure 6.** Combination of PDLIM2 nanotherapy, chemotherapy and immunotherapy shows great efficacy in lung cancer treatment. (**A**) IHC staining showing PD-L1 induction by chemotherapy but not PDLIM2 nanotherapy. (**B**) Schedule of lung cancer induction and treatment. (**C**) Urethane model showing high resistance of lung tumors to the chemo and αPD-1 combination therapy in lung epithelial specific PDLIM2 deletion mice (ΔSPC) (n≥4). (**D**) IHC staining showing no MHC-I induction by chemotherapy, PD-1 immunotherapy or their combination (n=6). (**E**) Tumor examination showing complete remission of all lung tumors in 60% of mice by combination of the three therapies (n≥5). (**F**) IHC staining showing increased TILs by PDLIM2 nanotherapy in mice treated with anti-PD-1 and chemotherapeutic drugs (n=6). (**G**) FACS analysis showing increased lung CD8+ T-cell activation by PDLIM2 nanotherapy in mice treated with anti-PD-1 and chemotherapeutic drugs (n=5). (**H**) No significant effect of PDLIM2 nanotherapy on the body weight of mice treated with anti-PD-1 and chemotherapeutic drugs (n=5). Scale bar in (**A and F**), 20 μm. Student's *t* test was performed (two tailed, unpaired) and data represent means ± SEM in (**c–g**). *p<0.05; **p<0.01; ns, not statistically significant.

The online version of this article includes the following source data and figure supplement(s) for figure 6:

**Figure supplement 1.** PDLIM2 nanotherapy causes no obvious toxicity in major organs.

**Figure supplement 2.** Epigenetic drugs cause body weight loss in mice with lung cancer.

**Figure supplement 2—source data 1.** Excel file for the data shown in *Figure 6—figure supplement 2*.

**Figure supplement 3.** Epigenetic drugs show better efficacy in lung cancer treatment.

**Figure supplement 3—source data 1.** Excel file for the data shown in *Figure 6—figure supplement 3*.

**Source data 1.** Excel file for the data shown in *Figure 6C-H*.

chemotherapy cannot induce MHC-I expression, which limits further improvement of its synergy with immune therapy for complete cancer remission. Also, tumor cells usually express high levels of survival genes against the tumoricidal effects of chemotherapeutic drugs and of CTLs, including those activated by chemotherapy and unleashed by ICIs.

On the other hand, PDLIM2 nanotherapy induces MHC-I expression and lymphocyte tumor infiltration but does not up-regulate PD-L1. Moreover, PDLIM2 nanotherapy prevents the induction of MDR1 and the expression of tumor-related genes and in particular cell survival genes, further sensitizing tumor cells to the cytotoxicity of chemotherapeutic drugs and immune cells including those recruited by chemotherapy and unleashed by PD-1 blockade. Because of these important functions of PDLIM2 nanotherapy, PDLIM2 nanotherapy improved the efficacy of chemotherapy and PD-1 blockade therapy, and in combination with chemotherapy and PD-1 blockade therapy, resulted in complete cancer remission in most of the animals and dramatic tumor reduction in the remaining mice.

Another important clinical characteristic of PDLIM2 nanotherapy is its tumor-specificity and high safety profile. It delivered PDLIM2-expression plasmids to lung tumor tissues and showed undetectable toxicity in the animal model. Its combination does not further increase the toxicity of anti-PD-1 and chemotherapeutic drugs. This is in sharp contrast to the Food and Drug Administration (FDA)-approved epigenetic drugs, which can restore PDLIM2 expression in cancer cells with PDLIM2 epigenetic repression (*Sun et al., 2019*; *Guo and Qu, 2021*; *Qu et al., 2010a*; *Qu et al., 2010b*; *Sun et al., 2015*; *Yan et al., 2009b*; *Vanoirbeek et al., 2014*). Epigenetic drug treatment leads to body weight loss of animals (*Figure 6—figure supplement 2A*). The toxicity is much worse when epigenetic drugs are combined with chemotherapy (*Figure 6—figure supplement 2*).

Besides its side effects, epigenetic therapy should have much more limited application, even though it shows better efficacy as a monotherapy or combined with chemotherapeutic drugs or anti-PD-1 in comparison to PDLIM2 nanotherapy (*Figure 6—figure supplement 3*). While nanoPDLIM2-based combination therapies could be applicable to all lung tumors with PDLIM2 repression regardless of the mechanisms involved, epigenetic therapy may be used to treat about 26% of lung tumors with PDLIM2 epigenetic repression only. About 58% of lung tumors harboring PDLIM2 LOH are not suitable to epigenetic therapies, although most of them are also with epigenetic alterations of the *pdlim2* gene. Of note, the therapeutic efficacy of epigenetic therapy depends on PDLIM2 expression (*Sun et al., 2019*), and PDLIM2 heterozygous loss causes spontaneous lung and other cancers. Further, the efficacy and toxicity of the combination of epigenetic agents with both anti-PD-1 and chemotherapeutic drugs has not been examined yet.

In summary, the presented data identify genetic deletions as a major mechanism other than epigenetic alterations for PDLIM2 repression in human lung cancer, and PDLIM2 as a haploinsufficient tumor suppressor particularly important for suppressing lung cancer and therapy resistance. More importantly, these preclinical data establish a novel combination treatment of nanoPDLIM2, anti-PD-1 and chemotherapeutic drugs that induces complete remission of all lung tumors in most animals and is also with high safety profile. We believe that these knowledges are applicable to other cancers, because PDLIM2 repression has also been linked to numerous human cancers other than lung cancer.

## Materials and methods

**Key resources table**

| Reagent type (species) or resource | Designation | Source or reference | Identifiers | Additional information |
|---|---|---|---|---|
| Gene (*Homo sapiens*) | *PDLIM2* | GenBank | 64236 | |
| Gene (*Mus musculus*) | *Pdlim2* | GenBank | 213019 | |
| Strain, strain background (*Mus musculus*) | FVB/N | Ref# 6, 28 | *Pdlim2*flx/flx/*SP*-C-rtTAtg/-/(tetO)7CMV-Cretg/tg (ΔSPC) | |
| Strain, strain background (*Mus musculus*) | BAB/c | Ref# 6, 19, 21, 22 | *Pdlim2*-/- | |
| Commercial assay or kit | in vivo-jetPEI | Polyplus Transfection | 101000030 | |

*Continued on next page*

*Continued*

| Reagent type (species) or resource | Designation | Source or reference | Identifiers | Additional information |
|---|---|---|---|---|
| Recombinant DNA reagent | pCMV-myc-Pdlim2 | This paper | PDLIM2 plasmid | PDLIM2 expression plasmid |
| Chemical compound, drug | Carboplatin | AdipoGen | AG-CR1-3591 | |
| Chemical compound, drug | Paclitaxel | AdipoGen | AG-CN2-0045 | |
| Cell line (*Homo sapiens*) | Lung cancer cell lines | Ref# 6 | Calu-6 (RRID:CVCL_0236) H727 (RRID:CVCL_1584) H23 (RRID:CVCL_1547) H358 (RRID:CVCL_1559) SKLU-1 (RRID:CVCL_0629) SW1573 (RRID:CVCL_1720) CALU-1 (RRID:CVCL_0608) 128-88 T (RRID:CVCL_A2AG) H1299 (RRID:CVCL_0060) 273T (RRID:CVCL_Y296) HCC827 (RRID:CVCL_2063) H1650 (RRID:CVCL_1483) H3255 (RRID:CVCL_6831) 343T (RRID:CVCL_A2AK) Calu-3 (RRID:CVCL_0609) H1435 (RRID:CVCL_1470) H1793 (RRID:CVCL_1496) H596 (RRID:CVCL_1571) H838 (RRID:CVCL_1594) H1838 (RRID:CVCL_1594) A-549 (RRID:CVCL_0023) H1975 (RRID:CVCL_1511) | Cell lines maintained in the laboratories of Dr. Gutian Xiao and Dr Zhaoxia Qu |
| Cell line (*Homo sapiens*) | nontumorigenic bronchial epithelial cell line from normal adult | ATCC | NL20 (RRID:, CVCL_3756, ATCC# CRL-2503) | |
| Software | Flow data acquisition and analysis | BD Biosciences | Accuri C6 | |
| Software | Flow data acquisition | BD Biosciences | FACSDiva | |
| Software | Flow data analysis | FlowJo | FlowJo | |
| Software | Data statistical analysis and graph presentation | GraphPad | Graphpad Prism | |

## Animals and lung carcinogenesis

We have complied with all relevant ethical regulations for animal testing and research. The animal experiments were performed in accordance with the US National Institutes of Health (NIH) Guidelines on the Use of Laboratory Animals. All animals were maintained under pathogen-free conditions and used according to protocols approved by the Institutional Animal Care and Use Committee (IACUC) of the University of Pittsburgh. *Pdlim2*flx/flx/*SP-C*-rtTA$^{tg/-}$/(tetO)7CMV-Cre$^{tg/tg}$ (ΔSPC) mice under a pure FVB/N background and *Pdlim2*$^{-/-}$ mice under a pure BALB/c background have been described before (*Sun et al., 2019*; *Tanaka et al., 2005*; *Tanaka et al., 2007*; *Qu et al., 2012*; *Li et al., 2021*). For lung carcinogenesis, 6- to 8-week-old ΔSPC and wild type FVB/N mice were intraperitoneally (i.p.) injected with urethane (1 mg/g body weight, Sigma-Aldrich, St. Louis, MO, USA) once a week for 6 consecutive weeks (*Sun et al., 2019*), followed by different treatments as shown in the figures. Mice were euthanized at six weeks post urethane treatment for examination of lung tumors, inflammation, and treatment-induced toxicity. Surface tumors in mouse lungs were counted blinded under a dissecting microscope, and tumor diameters were measured by microcalipers.

## Cell lines

All the human lung cancer cell lines were originally obtained from colleagues in University of Pittsburgh (*Sun et al., 2019*) and maintained in the lab. The nontumorigenic human bronchial epithelial

cell line NL-20 was originally purchased from American Type Culture Collection (ATCC, Cat# CRL-2503). The cell lines were authenticated by short tandem repeat profiling, and have been tested negative for Mycoplasma.

## Preparation of PDLIM2-expression plasmid or empty vector plasmid nanoparticles

The polyethylenimine (PEI)-based nanoparticles (in vivo-jetPEI) (Polyplus Transfection, New York, NY, USA) and plasmid DNA complexes at a nitrogen-to-phosphate ratio of 8 (N/$P$=8) were prepared according to the manufacturer's instructions. Briefly, 25 µg of pCMV-myc-*Pdlim2* or empty vector plasmids in 100 µl of a 5% glucose solution were mixed with the in vivo-jetPEI reagent (4 µl) diluted into 100 µl of a 5% glucose solution. After 15 min of incubation at room temperature, the mixed solution (200 µl/mouse) was injected intravenously (i.v.) via the tail vein. PDLIM2-expression plasmid or empty vector plasmid DNA in tissues were measured by quantitative PCR assays targeting *Amp-R* in the genome of these plasmids, normalized with *Lyz2* within the mouse genome.

## Histology and immunohistochemistry (IHC) analysis

Lung, liver, kidney, and spleen tissues were excised, fixed in formalin, embedded in paraffin, and cut into 4-µm-thick sections. Sections were stained with H&E or subjected to sequential incubations with the indicated primary antibodies, biotinylated secondary antibodies and streptavidin-HRP (*Sun et al., 2019*). Antibodies were listed in *Supplementary file 1a*.

## In vivo BrdU labeling

Mice were i.p. injected with 50 mg/kg BrdU (Sigma-Aldrich, St. Louis, MO, USA) 24 hr prior to euthanasia. Mouse lung tissue sections were stained with anti-BrdU (Sigma-Aldrich, St. Louis, MO, USA). BrdU labeling index was calculated as the percentage of labeled cells per total cells counted (>500 cells in each counted tumor-containing area).

## Flow cytometry (FACS) analysis

The cells were isolated from mouse lungs using collagenase/dispase digestion followed by filtration with 70 µm Nylon cell strainer and red blood cell lysis; then the cells were incubated with the antibodies against cell surface antigens after blocked with αCD16/CD32. The cells were then fixed with paraformaldehyde (2%), permeabilized and incubated with antibodies against intracellular antigens if needed. For interferon-γ (IFNγ) staining, cells were treated with phorbol 12-myristate 13-acetate (PMA, 50 ng/ml), ionomycin (1 µM), brefeldin A (BFA, 3 µg/ml), and monensin (2 µM) for 4 hr before they were stained for FACS analysis. Data were acquired and analyzed by Accuri C6 or LSRFortessa I (BD Biosciences) and FlowJo software (*Sun et al., 2021*).

## Quantitative polymerase chain reaction (qPCR) analysis

The indicated tissues or cells were subjected to DNA or RNA extraction, RNA reverse transcription and real-time PCR using trizol, reverse transcriptase, and Power SYBR Green PCR Master Mix (Thermo Fisher Scientific, Waltham, MA, USA) according to manufacturer's protocol. Primer pairs used for qPCR were listed in *Supplementary file 1b*.

## Microsatellite and gene-specific PCR-based LOH analysis of *PDLIM2*

Genomic DNAs were isolated from human lung tumors and their matched normal tissues using the PureLink Genomic DNA Purification Kit (Invitrogen, Carlsbad, CA, USA), and subjected to semi-quantitative PCR using the primers specific for the microsatellite markers D8S1786 and D8S1752 that straddle the *PDLIM2* genetic locus or the *PDLIM2* genetic locus itself. Primer pairs used for the assays were listed in *Supplementary file 1b*.

## Statistical analysis

One-way ANOVA power analysis was used to determine the minimum sample size. Animals were randomly assigned to different treatment groups. Measurements were taken from distinct samples. Student's t test (two tailed) and one-way ANOVA/Tukey's or two-way ANOVA/Sidak's test were used to assess significance of differences between two groups and multiple comparisons,

respectively. Pearson's correlation test was used to assess association between *PDLIM2* expression with its promoter methylation or genetic deletion and the overlap between *PDLIM2* promoter methylation and genetic deletion. All bars in figures represent means ± SEM. The *p* values are indicated as *p < 0.05, **p < 0.01, ns, not statistically significant, except for those shown in figures. The p-values < 0.05 and 0.01 are considered statistically significant and highly statistically significant, respectively. Grubbs' and ROUT outlier tests were established a priori. There were no exclusions in the analyses.

## Acknowledgements

The authors thank Dr. MJ Grusby (Harvard School of Public Health) and Dr. JA Whitsett (University of Cincinnati College of Medicine) for providing PDLIM2$^{-/-}$ and SP-C-rtTA$^{tg/-}$/(tetO)7CMV-Cre$^{tg/tg}$ mice, respectively. The authors also thank Dr. W Ma and Dr. S Li (University of Pittsburgh) for their suggestions on the nanoparticle preparation, and Dr. LH Rigatti (University of Pittsburgh) for her histological diagnosis of mouse tissues. This study was financially supported in part by the NIH National Institute of General Medical Sciences (NIGMS) grant R01 GM144890, National Cancer Institute (NCI) grants R01 CA172090 and R01 CA258614, R21 CA259706, American Cancer Society (ACS) Research Scholar grant RSG-19-166-01-TBG, American Lung Association (ALA) Lung Cancer Discovery Award 821321, and Tobacco Related-Disease Research Program (TRDRP) Research Award T33IR6461.

## Additional information

### Funding

| Funder | Grant reference number | Author |
|---|---|---|
| National Institutes of Health | R01 GM144890 | Gutian Xiao Zhaoxia Qu |
| National Institutes of Health | R01 CA172090 | Gutian Xiao |
| National Institutes of Health | R01 CA258614 | Gutian Xiao Zhaoxia Qu |
| National Institutes of Health | R21 CA259706 | Gutian Xiao |
| American Cancer Society | Research Scholar Grant RSG-19-166-01-TBG | Zhaoxia Qu |
| American Lung Association | Lung Cancer Discovery Award 821321 | Zhaoxia Qu |
| Tobacco-Related Disease Research Program | Research Award T33IR6461 | Zhaoxia Qu |

The funders had no role in study design, data collection and interpretation, or the decision to submit the work for publication.

### Author contributions

Fan Sun, Pengrong Yan, Data curation, Formal analysis, Validation, Investigation, Visualization, Methodology, Writing – review and editing; Yadong Xiao, Data curation, Validation, Investigation, Visualization, Methodology, Writing – review and editing; Hongqiao Zhang, Writing – review and editing; Steven D Shapiro, Conceptualization, Supervision, Investigation, Methodology, Writing – review and editing; Gutian Xiao, Zhaoxia Qu, Conceptualization, Resources, Data curation, Formal analysis, Supervision, Funding acquisition, Validation, Investigation, Visualization, Methodology, Writing – original draft, Project administration, Writing – review and editing

### Author ORCIDs

Hongqiao Zhang  https://orcid.org/0000-0002-0526-5636
Zhaoxia Qu  https://orcid.org/0000-0002-2769-9814

## Ethics

We have complied with all relevant ethical regulations for animal testing and research. The animal experiments were performed in accordance with the US National Institutes of Health (NIH) Guidelines on the Use of Laboratory Animals. All animals were maintained under pathogen-free conditions and used according to protocols approved by the Institutional Animal Care and Use Committee (IACUC) of the University of Pittsburgh (Animal Welfare Assurance Number D16-00118).

Reviewer #1 (Public Review): https://doi.org/10.7554/eLife.89638.3.sa1
Reviewer #2 (Public Review): https://doi.org/10.7554/eLife.89638.3.sa2
Author response https://doi.org/10.7554/eLife.89638.3.sa3

## Additional files

### Supplementary files
• MDAR checklist
• Supplementary file 1. Antibodies and primers used.

### Data availability

TCGA lung adenocarcinoma, lung squamous cell carcinoma, and lung cancer data we analyzed were obtained from the Cancer Genome Atlas Program (https://www.cancer.gov/tcga). All data generated or analyzed during this study are included in the manuscript and supporting files; source data files have been provided for all the figures.

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
