## [Editor Report · eLife assessment]

This study presents a **valuable** finding for the immunotherapy of cancer. The data support the role of PDLIM2 as a tumor suppressor, and more immediately, its relevance for strategies to improve the efficacy of immunotherapy. The evidence supporting the conclusions is **compelling** and the work will be of interest to biomedical scientists working on cancer immunology.

---

## [Referee Report · Reviewer #1 (Public Review)]

The manuscript by Sun and colleagues followed on their previous findings on the tumor suppressive role of PDLIM2 in lung cancer. They further investigated various mechanisms, including epigenetic modification, copy number variation and LOH, that led to the decrease expression of PDLIM2 in human lung cancer. Next, they used nanoparticle-based approach to specifically restore the expression in mouse lung tumors. They showed that over-expression PDLIM2 in lung cancer repressed its progression in vivo. Also, this treatment could synergize with chemotherapy and checkpoint inhibitor anti-PD-1. Overall, the results were quite promising and convincing, using a treatment combination that would appear to have potential for clinical implementation.

---

## [Referee Report · Reviewer #2 (Public Review)]

Summary: The authors have previously demonstrated that the E3 ligase PDLIM2 inhibits NF-kB and STAT3 and is epigenetically repressed in human lung cancers (Sun et al. Nat. Comm. 2019 10: 5324); therefore, PDLIM2 is a tumor suppressor in lung cancer. In this manuscript, they follow up on their previous findings and show that expression of PDLIM2 is downregulated in human lung cancers by both genetic deletion and promoter methylation. They further describe a novel approach to restore the expression of PDLIM2 in mouse lung tumors by systemically administering PDLIM2 plasmids encapsulated in nanoparticles (termed "nanoPDLIM2"). The nanoPDLIM2 approach was shown to exhibit efficacy with low toxicity in a urethane-induced mouse lung cancer model. The authors further demonstrated synergy of nanoPDLIM2 with chemotherapy and PD-1 blockade immunotherapy. The combination therapy of nanoPDLIM2, chemotherapy and immunotherapy proved most effective with complete tumor remission in 60% of mice. Mechanistically, nanoPDLIM2 upregulated MHC-I expression, enhanced CD4/CD8 T cell activation and tumor infiltration, and suppressed MDR1 induction and nuclear expression of STAT3, RelA and prosurvival genes in tumors. Overall, this study is important because it reinforces the critical roles of PDLIM2 in suppressing lung cancer, and also identifies a potential approach to restoring PDLIM2 expression in lung tumors. The experiments were well executed; the data are convincing and support the conclusions made by the authors.

---

## [Author Response]

**Reviewer #3 (Public Review):**
Strengths:NanoPDLIM2, nanotechnologies that efficiently deliver lentivirus overcomes resistance to chemotherapy and anti-PD-1 immunotherapy. This is a new strategy for enhancing the efficiency of immune checkpoint inhibitors.This finding is important from a clinical translation perspective, but I have several minor concerns.Weaknesses:1. Please describe the mechanism of increased MHC class I and PD-L1 by PDLIM2.

Our previous studies showed that PDLIM2 induces MHC-I induction through decreasing STAT3 whereas it is dispensable for PD-L1 expression (Sun et al, 2019, PMID: 31757943). In line with the studies, PD-L1 is induced by chemotherapeutic drugs, but not by NanoPDLIM2 (Figure 6A). Together with the roles of PDLIM2 in repressing RelA-dependent MDR1 induction by chemotherapy and in preventing expression of cell survival and proliferation genes by targeting both RelA and STAT3 (Sun et al, 2019, PMID: 31757943), further providing the mechanistic basis for the combination and synergistic effect of nanoPDLIM2, anti-PD-1 and chemo drugs. The improvement has now been further incorporated.

1. Please describe the mechanism of decreased MDR1, nuclear RelA and STAT3 by PDLIM2.

Our previous studies demonstrated that PDLIM2 reduces MDR1 expression by degrading nuclear RelA (Sun et al, 2019, PMID: 31757943).

1. Please determine whether PDLIM2 expression directly impacts immune cells (function and number)?

As shown in Figure 5, NanoPDLIM2 increased the number and activation of tumor infiltrating lymphocytes (TILs); and in prior study, PDLIM2 knockout repressed the numbers of TILs and inhibited the activation of CD4+ and CD8+ T cells, while its re-expression in lung tumors led to T cell activation (Sun et al. 2019, PMID: 31757943). On the other hand, selective deletion of PDLIM2 in immune cells and in particular myeloid cells repressed the numbers and activation of TILs (Li et al, 2021, PMID: 33539325; PMCID: PMC8021114). Thus, PDLIM2 may impact immune cells both directly and indirectly, particularly when nanoparticles can deliver PDLIM2 into both tumor cells and tumor-associated immune cells (despite PDLIM2 is delivered into much fewer immune cells compared to tumor cells).

1. What is the efficiency of PDLIM2 delivery? Does delivery efficiency determine anti-tumor effect?

As shown in the manuscript, the dose of PDLIM2 used already shows high delivery (20-30 copies per tumor cell in Figure 3B) and therapeutic efficacy in the mouse model of refractory lung cancer and particularly when being combined with anti-PD-1 and chemo drugs. It is of interest to test different doses in the model for the best delivery and efficacy, which is actively being pursued in the lab.

1. Authors used a non-immunogenic tumor model. Can you demonstrate the combination effect with PDLIM2 in immunogenic lung cancer models to determine whether the combination of PDLIM2 with anti-PD-1 Ab confers a synergistic effect without chemotherapy?

Yes, it is of interest to demonstrate the combination of PDLIM2 and anti-PD-1 in immunogenic lung cancer models with chemotherapy although a synergy is highly expected. The greatest challenge in the lung cancer field is the low response of non-immunogenic tumor, which is the focus of the current manuscript.

1. On page 11, % change can make one over-interpret data.

The % change has been removed from the manuscript.

1. In Figure 5, what is the difference between 5A and 5D?

Figure 5A shows the increase of TILs by nanoPDLIM2 in animals that did not receive PD-1 blockade immunotherapy, Figure 5D shows the increase of TILs by nanoPDLIM2 in animals received PD-1 blockade immunotherapy.

1. It is unclear whether PDLIM2 confers an additive or a synergistic effect with anti-PD-1/chemo.

PDLIM2 nanotherapy confers a synergistic effect with chemotherapy on increasing apoptosis in tumors (Figure 4B) and tumor reduction (Figure 4A and 6E, left panel, tumor number), confers a synergistic effect with antiPD-1 on increasing CD4+ and CD8+ TILs (Figure 5A and 5D), and apoptosis in tumors (Figure 5F), and an additive effect on tumor reduction (Figure 5C and 6E), and confers a synergistic effect with chemotherapy plus anti-PD-1 on increasing CD4+ and CD8+ TILs (Figure 5A and 6F) and tumor reduction (Figure 6E, left panel, tumor number).

1. Have the authors tested any toxicity in normal lungs?

Same to tumor lungs, no obvious toxicity has been observed in normal lungs.

**Reviewer #1 (Recommendations For The Authors):**
The paper is clear and well-written, although some minor edits are needed. For example, the title could be changed to reflect both human and mouse studies in the manuscript for more general readers. Moreover, 'lung cancer' should be used instead of 'lung cancers'. The manuscript could be further improved by validating their findings in a different model and particularly the syngeneic model of metastatic lung cancer for a better overall survival time by the new combination therapy, given the fact that clinical trial studies usually start in patients with metastatic tumors. But this is optional because the therapeutic effect on primary lung cancer is already significant.

Thanks for the correction and wonderful suggestions. The “lung cancers” were replaced with “lung cancer”, and the title was changed to “Improving PD-1 blockade plus chemotherapy for complete remission of lung cancer by nanoPDLIM2”.

**Reviewer #2 (Recommendations For The Authors):**
1. What is the rationale for i.v. injection of nanoparticles containing PDLIM2 plasmid? Intranasal administration of nanoparticles may potentially target nanoPDLIM2 specifically to the lungs. Another potential option is intranasal infection of mice with adenovirus expressing PDLIM2.

The rationale for i.v. injection of nanoPDLIM2 is that iv injected nanoPDLIM2 first reach into the lung and more importantly tumor tissues as well as the convenience and high efficacy of mouse i.v. injection, particularly when multiple injections are needed. Mice are much less stressful compared to other intranasal or even intratracheal injection. Adenovirus can be used only once, because it will initiate ant-viral immune response in mice.

1. The authors examine PDLIM2 expression in lung tumors 1 week after i.v. administration of nanoparticles (Fig. 3A). Do all tumor cells express PDLIM2 after nanoPDLIM2 treatment? How long does PDLIM2 persist in the tumors? The kinetics of PDLIM2 expression may be informative to help interpret the results from the various combination treatments given to the mice. Multiple rounds of nanoPDLIM2 treatment could potentially improve the efficacy of the treatment.

For all the sections examined (n=6), PDLIM2 was re-expressed in most but not all lung cancer cells at 1-week of the i.v administration. Accordingly, nanoPDLIM2 was injected weekly. We are examining if PDLIM2 reexpression can last longer. We are also testing the best dose with the best efficacy.

1. Does the plasmid DNA from nanoparticles trigger an innate immune response in the lung that contributes to anti-tumor responses?

In line with previous studies showing no effect on immune responses (Bonnet et al. 2008. PMID: 18709489), the dose used in current study does not significantly affect immune cells in the lung, suggesting no obvious effect of nanoparticles with empty plasmid on innate immune response.

1. In Fig. 4, does the combination of nanoPDLIM2 and chemotherapy diminish STAT3 nuclear staining?

NanoPDLIM2 alone decreased nuclear STAT 3 in tumor cells (Figure 2C), it also diminished nuclear STAT3 in tumor cells with the combination of chemotherapy.